# Genome-wide diversity and global migration patterns in dromedaries follow ancient caravan routes

Sara Lado[1], Jean Pierre Elbers[1], Angela Doskocil[1], Davide Scaglione [2], Emiliano Trucchi[3], Mohammad Hossein Banabazi[4], Faisal Almathen[5,6], Naruya Saitou[7], Elena Ciani[8 ✉] & Pamela Anna Burger [1 ✉]

Dromedaries have been essential for the prosperity of civilizations in arid environments and the dispersal of humans, goods and cultures along ancient, cross-continental trading routes. With increasing desertification their importance as livestock species is rising rapidly, but little is known about their genome-wide diversity and demographic history. As previous studies using few nuclear markers found weak phylogeographic structure, here we detected fine-scale population differentiation in dromedaries across Asia and Africa by adopting a genome-wide approach. Global patterns of effective migration rates revealed pathways of dispersal after domestication, following historic caravan routes like the Silk and Incense Roads. Our results show that a Pleistocene bottleneck and Medieval expansions during the rise of the Ottoman empire have shaped genome-wide diversity in modern dromedaries. By understanding subtle population structure we recognize the value of small, locally adapted populations and appeal for securing genomic diversity for a sustainable utilization of this key desert species.

[1] Research Institute of Wildlife Ecology, Department of Interdisciplinary Life Sciences, University of Veterinary Medicine Vienna, Savoyenstrasse 1, 1160 Vienna, Austria. [2] IGA Technology Services, Via Jacopo Linussio, 51, 33100 Udine, Italy. [3] Department of Life and Environmental Sciences, Marche Polytechnic University, Via Brecce Bianche, 60131 Ancona, Italy. [4] Department of Biotechnology, Animal Science Research Institute of Iran (ASRI), Agricultural Research, Education & Extension Organization (AREEO), Karaj 3146618361, Iran. [5] Department of Veterinary Public Health, College of Veterinary Medicine, King Faisal University, Al-Hasa, Saudi Arabia. [6] The Camel Research Center, King Faisal University, Al-Hasa, Saudi Arabia. [7] Population Genetics Laboratory, National Institute of Genetics, 1111 Yata, Mishima 411-8540, Japan. [8] Department of Biosciences, Biotechnologies and Biopharmaceutics, University of Bari Aldo Moro, Via Orabona, 4, 70125 Bari, Italy. ✉email: Elena.Ciani@uniba.it; Pamela.Burger@vetmeduni.ac.at

As one of the most recently domesticated animals (~3000–4000 years before present; ybp), the dromedary (*Camelus dromedarius*) has a special position in human migration and trading[1]. Its physiological adaptations to harsh and dry environments allowed humans to traverse hostile lands such as deserts like no other livestock, including the horse[2]. Reversing a historian's observation "the sea unites rather than divides"[3] to the desert, dromedaries facilitated the expansion of civilizations[1,4,5]. Their superior and unique features predestined dromedaries for the use as military animals, and for the advance of international trading along ancient caravan routes, such as the Silk Road and Incense routes[6,7]. Interspecific hybridization between dromedaries and the closely related two-humped Bactrian camels (*Camelus bactrianus*) produced even more robust and enduring animals with an aptitude for the extreme climatic conditions of the Silk Road[8,9]. Nowadays, first generation hybrids and their back-crosses are valued in many countries for increased milk or wool production[10], as well as in famous camel wrestling events[11]. These remarkable commercial networks in human history facilitated domestic animal exchange across large geographical distances and acted as gene-flow corridors, not only for camels but also for other livestock[12]. While camels were the chosen animals for transportation, horse movements along the complete network of the Silk Road mainly occurred in forms of tributes and gifts[12,13].

The early progenitors (*Protylopus*) of camelids emerged in the North American savannah during the Eocence (~45 Mya). After their split into New (Lamini) and Old World (Camelini) camels around 15 Mya, the ancestors of the Old World camels crossed into Eurasia via the Bering Land Bridge (~6.6 Mya) and further diverged into one- and two-humped camels (reviewed in Burger et al.[14]). Early-domestic dromedaries (second and first millennium before Common Era; BCE) cohabited the coastal Southeast of the Arabian Peninsula for nearly one millennium together with wild specimens, which did not survive the beginning of the Common Era (CE)[15–19]. The early dispersal of modern dromedaries from the Arabian Peninsula to the Levant, North Africa, South Asia, and finally to Australia (introduced in the late 19th century)[20] was followed by cross-continental back-and-forth movements along historic trading routes. This led to a blurring of genetic stocks[21] culminating in a panmictic dromedary population at the mitochondrial DNA level[1]. Previous studies using a limited number of microsatellite markers and mtDNA detected only weak population structure in the global dromedary population[1,20,22]. As few genomic studies have been completed with the dromedary, this major ungulate species has been left out of the livestock genomic revolution. However, two draft reference genomes at the scaffold level have been released[23,24], as well as a draft reference genome at the chromosome level[25], which will facilitate further genomic investigations.

We wanted to understand how human-induced migration patterns and historic demographic changes might have influenced population structure in the global dromedary population. We sequenced 22,721 SNP markers to overcome the limitations of previous studies using not more than 20 microsatellites. With a global dataset including samples spread over three continents, we describe effective migration rates of modern dromedaries that follow their human-driven dispersal along ancient trading routes[7]. Understanding subtle population structure, which has been shaped by past and recent demographic events, will help in recognizing the value of small populations and securing genomic diversity for a sustainable utilization of this important livestock species in a globally changing world.

## Results and discussion

We performed double-digest restriction site associated DNA (ddRAD) sequencing on 122 dromedary DNA samples from 18 countries (Supplementary Data 1) representative of the species distribution range. We included one Bactrian camel to test for potential interspecific hybridization, as this continues to be a widespread practice in Central Asia that might have started as early as pre-Roman times[11]. Higher numbers of reads mapping to the Bactrian camel were detected in three individuals from Iran and in six from Kazakhstan (see "Methods"), and we decided to remove these samples from downstream analysis due to potential introgression from Bactrian camel (Supplementary Data 2). After stringent filtering for genotype and individual missingness, minor allele frequency and relatedness, the final dataset consisted of 95 dromedaries and 22,721 SNPs present in at least 75% of the individuals.

**Moderate genome-wide diversity and low population structure**. With 22,721 SNPs, we estimated expected ($H_E = 0.27 \pm 0.17$; mean ± SD) and observed ($H_O = 0.25 \pm 0.17$) heterozygosities in the global dromedary population ($n_{pop} = 17$; $n_{ind} = 95$). Separating the samples according to their continental origins, both Asian ($n_{ind} = 49$, $H_E = 0.27 \pm 0.17/H_O = 0.25 \pm 0.17$) and African dromedaries ($n_{ind} = 46$, $H_E = 0.26 \pm 0.17/H_O = 0.25 \pm 0.18$) showed similar genomic diversity. The mean $H_E$ ($t = -2.2641$, df = 45,398, $P = 0.02$) and inbreeding coefficients ($t = -2.5159$, df = 43,024, $P = 0.01$) were higher in Asian than African dromedaries, but mean $H_O$ ($t = -1.2791$, df = 45,385, $P = 0.2$) was not different between continents, according to Welch's $t$ test. Complete diversity and inbreeding values are given in Supplementary Table 1. In comparison with other domestic species, i.e., sheep ($H_E = 0.22–0.32$)[26] or cattle ($H_E = 0.24–0.30$)[27], we consider the genome-wide diversity in dromedaries as moderate at the best. Several bottlenecks during the last glacial period (see demographic analysis below, and Fitak et al.[24]) and during domestication left modern dromedaries with a minimum of only six maternal lineages[1] and limited genome-wide diversity. This will have implications on future intensification of breeding and genomic selection in dromedaries from regions with increasing desertification.

In general, the genome-wide differentiation within the global dromedary population was very low. Analysis of Molecular Variance (AMOVA) revealed that most of the variation, ~94.3%, is explained within individuals (Supplementary Table 2). Allelic richness (AR) was similar between countries (AR = 0.25–0.27) with exception of Kenya which was lower (AR = 0.21). The pairwise fixation index between African and Asian individuals was very low ($F_{ST} = 0.006$; $P < 0.001$), and indices between dromedaries from different countries (if significant at all) were lowest in geographically close populations (e.g., Libya/Algeria: $F_{ST} = 0.0002$) and increased with geographic distance (Pakistan/Tunisia: $F_{ST} = 0.0328$) (Supplementary Table 3).

We screened for loci deviating from neutraliy using BayeScan 2.1[28] and identified sixteen $F_{ST}$ outliers to be putatively under selection (false discovery rate (FDR) <0.05) between African and Asian dromedaries (Supplementary Fig. 1). We found it reasonable to investigate the biological functions of those genes harboring the SNPs as they might relevant for the adaptation of dromedaries to their respective environments. We found SNPs in two genes, *CALN1* and *TREM1*, which are responsible for calcium ion binding and amplifying inflammatory responses triggered by bacterial and fungal infections, respectively (scaffold:SNP-location; JWIN01030783.1:128274 and JWIN01033764.1:729703). In addition, we examined (potentially linked) regions 200 kbp upstream and downstream of the $F_{ST}$-outlier loci and detected fifty-three genes related to a number of biological functions (Supplementary Data 3). Interestingly, around one fifth of the detected genes had functions related to the immune system

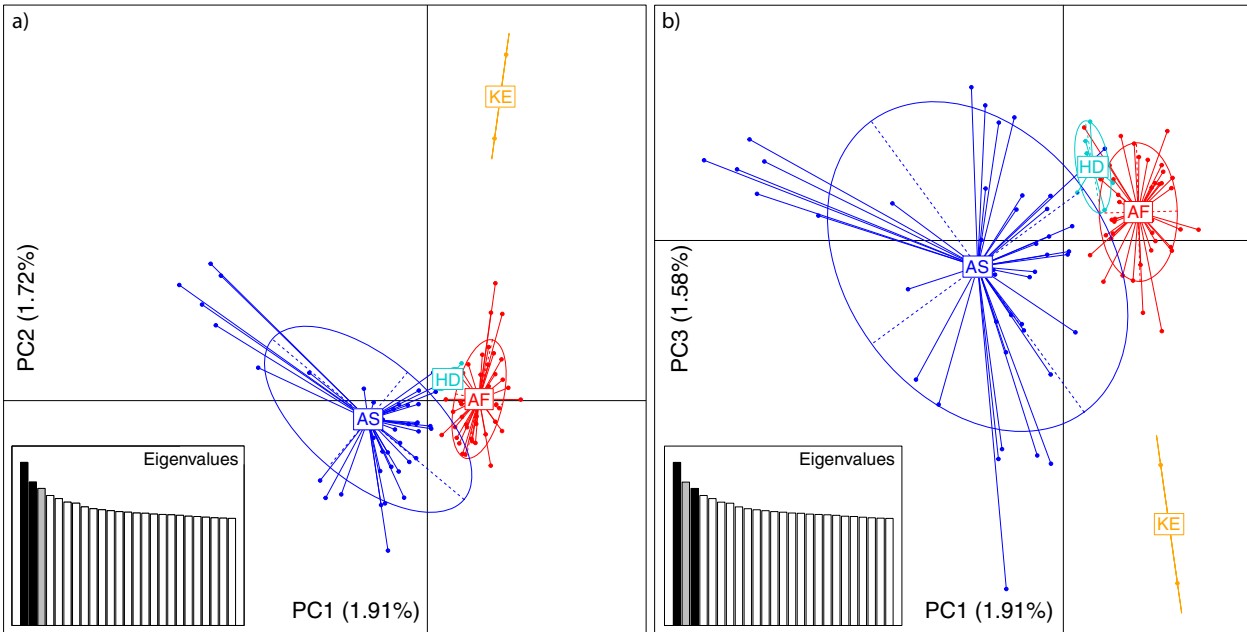

**Fig. 1 Population substructure in the global dromedary population.** Principal component (PC) analysis based on 22,721 SNPs to represent populations pre-classified according to phylogenetic clustering (Supplementary Fig. 2), Africa (AF), Asia (AS), *Hadhana* (HD), and Kenya (KE). Elipses refer to the distribution of individuals within groups. The first three PCs explain 5.3% of the total variation and are shown in **a** PC1–PC2 and **b** PC1–PC3.

hinting to an adaptive process in response to different pathogens in the respective environments. Other protein coding gene functions were related to pathways such as circadian rhythm, (ga)lactose, metabolism, reproductive or various cellular and developmental processes. A full list of genes is presented in Supplementary Data 3. Signatures of selection related to photoperiod, metabolism, immunity and growth have also been observed in chicken[29], sheep[30], and cattle (*TGFB3*[31]).

To understand the low genome-wide differentiation in dromedaries across their global range, we investigated population structure and admixture between populations. We projected the genetic variation of each dromedary on the first three axes inferred from a principal component analysis (PCA) and incorporated continental information (Africa/Asia) for each sample (Fig. 1). Principal component 1 (PC1) clearly separated African from Asian dromedaries, while PC2 and PC3 split Kenyan individuals from the rest of Africa and identified a single population from Saudi Arabia grouping closer to African than Asian individuals, although showing some cross-continental admixture (Fig. 1 and Supplementary Fig. 2). This separated population belongs to a specific breed, *Hadhana*, and is one of the twelve recognized dromedary ecotypes in Saudi Arabia, limited to mountain regions in the South of the Arabian Peninsula, Al-Baha[32]. In this case, the geographic accessibility might have an important role in the observed genetic distinctiveness. A possible explanation for the close relationship of *Hadhana* and African dromedaries might be the historic sea route from Jiddah in Saudi Arabia to Aydhab and Port Sudan. On the western coast of the Red Sea existed a trading route connecting the Horn of Africa to Petra and Damascus via Port Sudan, Aydhab and Myos Hormos, near today's Kosseir (Fig. 2)[6,7]. In general, the Asian dromedary population showed higher genetic variability, although the genetic variation explained by the three first axes was rather low with only 5.3% (Fig. 1). While this could be a sign for ancestral variation (the Arabian Peninsula was a center of domestication[1], we cannot discard the hyphothesis of post-domestication movements of camels or multiple origins of the founder populations as this would have left similar signals in the genomes.

We next inferred potential ancestry and admixture among Asian and African dromedaries using unsupervised genetic clustering in ADMIXTURE[33] (Fig. 3). Based on the lowest cross-validation error, the best clustering solution was 1 (Supplementary Fig. 3), which suggests a panmictic dromedary population and reflects the low genetic differentiation of 0.6% among individuals from different continents. Increasing the numbers of potential ancestral populations ($K$) from two to seven confirmed the already observed differentiation between African and Asian dromedaries ($K = 2$), the clustering of the Saudi Arabian *Hadhana* breed with Africa ($K = 4$), the separation of Kenyan and *Hadhana* individuals, and the higher number of distinct clusters on the Asian continent ($K = 7$). We find a more homogenous gene pool in African animals with the exception of the East African group[1] (Figs. 1, 2 and Supplementary Fig. 2), represented in our dataset by the two Kenyan dromedaries. This can be a consequence of a random founder effect followed by lack of gene flow due to geographical, physiological (e.g., Trypanosome infestation) and/or cultural barrier, i.e., dromedaries in East Africa were dominantly used for milk production rather than transport or riding[1]. There is a need to proceed with comprehensive analyses about the potential nature of natural and/or anthropogenic obstacles for gene flow between East African and other dromedaries.

**Effective migration rates along ancient caravan routes**. To formally test our qualitative observations of weak population structure among African and Asian dromedaries (Figs. 1, 3), we visualized the global spatial population structure using the Estimated Effective Migration Surfaces (EEMS) method[34]. Based on a stepping-stone model, EEMS detected a corridor of significantly higher effective migration rates than the overall mean along the Mediterranean coast, connecting the Northern parts of Africa and the Arabian Peninsula until the border of the Arabian Desert (Fig. 2). This pattern shows a continuous gene flow throughout the coastal dromedary populations, and a lower than average migration in the inland desert populations. A known trading route which fits this observed effective migration pattern

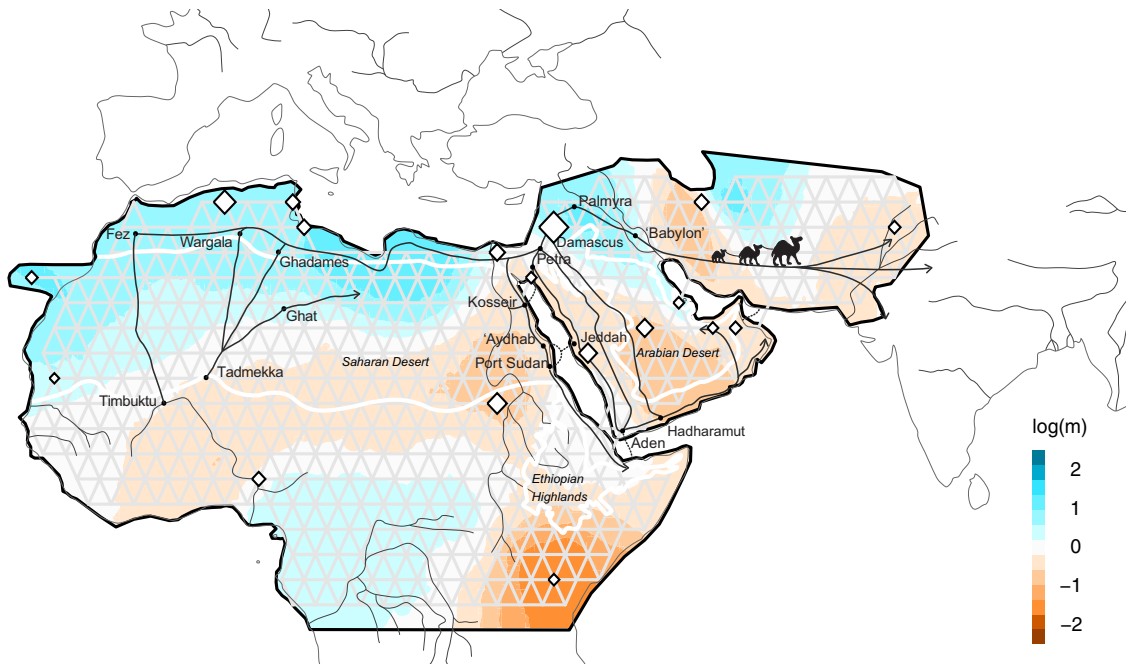

**Fig. 2 Estimated Effective Migration Surfaces (EEMS) in the global dromedary population.** EEMS plot representing the posterior mean of effective migration rates (m) (on a log10 scale) across space. With this normalization, significantly higher than the overall average rates are represented in blue ("corridors") and significantly lower than the overall average rate ("barriers") are represented in brown. Zero corresponds to the overall mean migration rate. Samples are represented by diamonds and the size is proportional to the number of sampling. Approximate coordinates are used. Ethiopian Highlands and Arabian Desert are highlighted with white lines. Black lines represent historical network of caravan routes, i.e., Incense and Silk roads[6,7] and main trans-Saharan gold trade networks[36] (adapted from[2]).

bordered the Mediterranean coast connecting Northwestern Africa to the North of the Arabian Peninsula from where caravans traveled toward Southern Asia along the Silk Road (Fig. 2)[1,7]. The introduction of the dromedary into Northern Africa via the Sinai from Roman Egypt started in the early first millennium BCE and intensified in the Ptolemaic period[6,15]. From there, dromedaries migrated along the Mediterranean coast, as archeological evidence dates their presence in Northwest Africa to the fourth to seventh century CE (Late Antiquity/Early Middle Ages)[1,6]. Even earlier dispersal of taurine cattle along the Northern coast of Africa and through the Mediterranean sea to Europe was described during the Bronze age[35].

It is clear that camels, unlike other domesticated species, were able to penetrate deep into the Saharan desert and to connect trans-Saharan cultures. West Sahara belonged to an Islamic trading network classified as one of the major gold suppliers in the ninth to tenth centuries CE[36] (Fig. 2). Tadmekka, a territory located in the Southwestern Saharan desert and governed by the Tuareg, was operational by the eighth century CE and was one of the earliest towns established in the region where cross-Saharan camel caravans traded[37]. These trades prolonged at least until the fourteenth century when Timbuktu, which similar to Tadmekka hosted large groups of Islamic traders, engaged in coin-based exchange economies across the Sahara[36].

While modern dromedaries along the western part of the Silk Road are still well connected today, the Iranian and Afghanistan deserts seem to present obstacles of effective migration. As EEMS assumes uniform migration rates the observed "barriers", could however be assessed as areas of lower population density with fewer migrants exchanged per generation, producing an effective "barrier" to gene flow[34]. The three main parallel itineraries of the Incense Routes through the Arabian Desert connecting the South of the Arabian Peninsula with the Levant[7] also showed lower than average migration rates (Fig. 2), which could be interpreted

as lower population density or potential sampling gaps. These trading routes were essential during historical periods, not only for exchanging luxury products (e.g., incense or gems), but for trading everyday local products[38].

The strongest barrier detected in our dataset concerned dromedaries from the Horn of Africa, which had the lowest genetic effective migration rates (Fig. 2). Geographical isolation due to the Ethiopian highlands, which might disrupt gene flow with northern populations, and the Golf of Aden would be the most likely explanation for the observed pattern. Genetic differentiation of livestock populations in East Africa has been described previously[1,22,39].

**Late Pleistocene population decline and medieval expansions.** To complete our understanding of the moderate genome-diversity observed in the global dromedaries, we investigated the demographic history and inferred effective population size ($N_E$) over time with an Extended Bayesian Skyline approach (EBS)[40] (Fig. 4). As genetic differentiation between Asian and African dromedary populations was low (0.6%) but highly significant ($P < 0.001$), we tested whether the dromedary populations from the two continents experienced a similar demographic history. First, we investigated the global population and second, African and Asian individuals separately. With the latter approach, we accounted for a potential confounding effect of population structure for the inference of $N_E$[40]. Due to the observed substructure in Kenyan and *Hadhana* individuals, we excluded these two populations from the continental groups.

Irrespective of the continental origin, all inferences showed similar patterns of an initial population expansion from one million ybp until ~700,000 ybp (Fig. 4). Our genome-wide population approach confirmed previous $N_E$ estimates based on a single dromedary whole genome sequence[23] using pairwise sequentially Markovian coalescent[41]. This population expansion

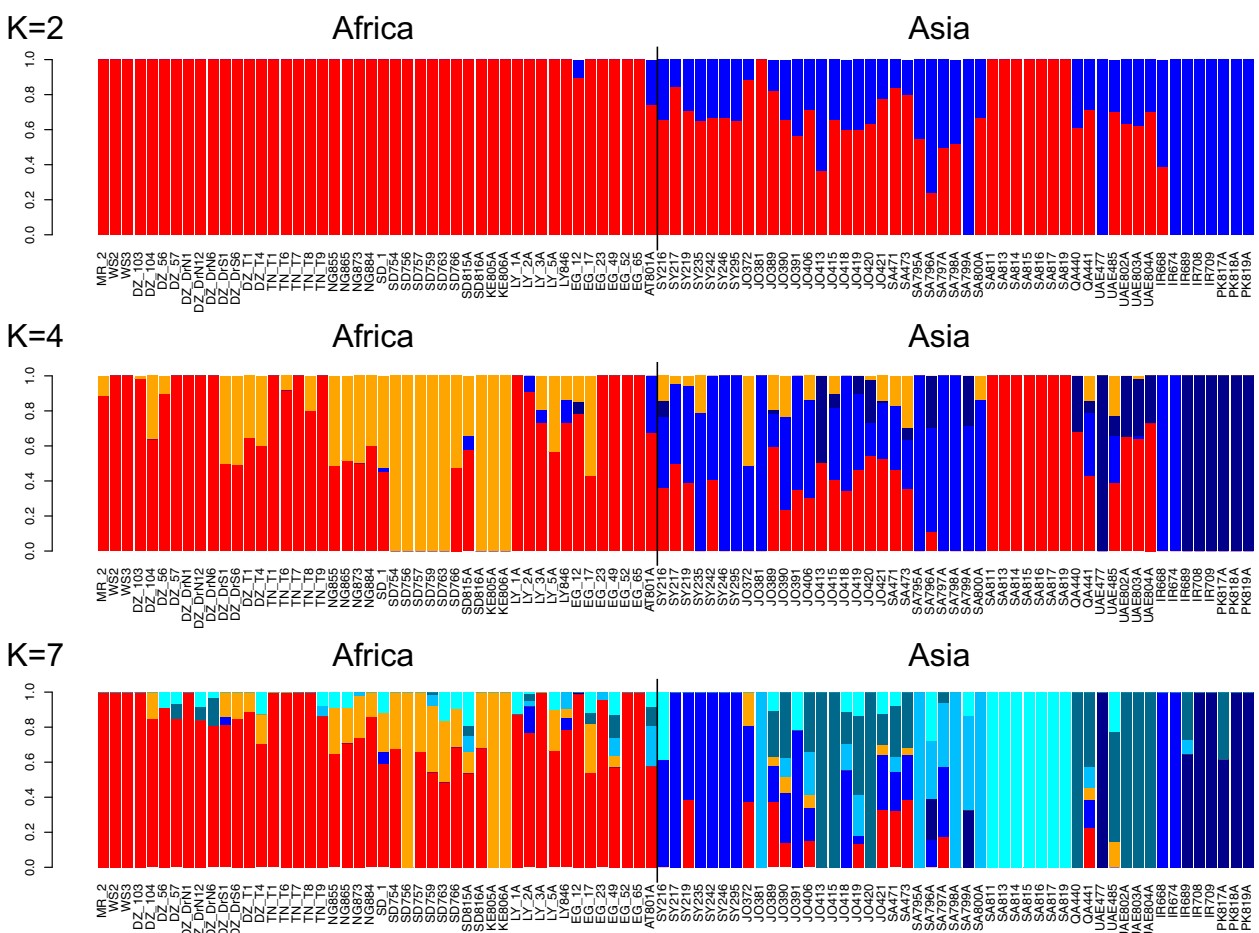

**Fig. 3 Admixture analysis of the global dromedary population.** Admixture analysis showing the proportion of potential ancestral populations ($K = 2$; 4; 7) for each individual (single bar). The geographical origins for each sample are represented below the bars in two-digit international country codes, where the middle line divides the first half (African countries) from the second half (Asian countries). Hadhana population is depict in cyan blue ($K = 7$).

coincides with two remarkable periods: the middle Pleistocene transition (1.25–0.70 million ybp) characterized by climatic cycles, and the Galerian Mammal Age (1.2–0.60 million ybp), which influenced the distribution and evolution of biota and resulted in some species being adapted to arid, cold climates[42,43]. Moreover, this timeframe also overlaps with the maximal diversity of the family Camelidae (early Galerian), supporting the adaptation of the dromedary ancestor to environmental changes with an expansion of its population during the middle Pleistocene transition[23,44]. Population expansion was followed by a drastic decline in $N_E$ beginning 700,000 ybp until the dromedary population collapsed during the last glacial period (LGP; 100,000–20,000 ybp)[45]. This is a finding shared by previous Old World[9] and New World camelid[23] studies and those focusing on Late Quaternary Megafauna[46].

Conversely, no bottleneck was picked up by any of the EBSPs during the time scale when dromedaries are predicted to have been first domesticated ~3000–4000 ybp[15,16]. Previous BSP analysis using mtDNA likewise did not show a population decline during the time of domestication[1]. It is possible that the detection of a bottleneck with the EBSP analysis related to domestication has been superimposed by the drastic decrease in $N_E$ ending ~30,000 ybp (Fig. 4). Similar demographic changes were observed in alpacas[23], where three population bottlenecks were detected throughout the cold conditions of the LGM in South America, yet no bottleneck was visible during the domestication period.

After the Pleistocene bottleneck, the dromedary population slowly increased until reaching a stable $N_E$ around 300 ybp, with a higher $N_E$ present within Asia than in Africa. Demographic inferences based on mtDNA sequences described slightly earlier expansion of the maternal lineages around 600 ypb[1] associated with the rise of the Ottoman empire and the conquest of Constantinople (1453 CE), followed by the extension to Southern Asia and the Red Sea coasts[47].

## Conclusions

Our study shows that assessing the evolutionary history of species using genome-wide approaches allows detailed inferences of population structure, migration, and potential signals of environmental adaptation. As the movements of dromedaries parallel those of humans, knowledge on dromedary spatial genetic signatures also sheds light into past human history. We detected genetic admixture across continental populations (Asia and Africa), which highlights the strong anthropogenic influence on these animals.

Human history is marked with the efforts for overcoming obstacles, be they of geographical (mountains, sea, and deserts) or cultural nature. Domestic animals, and in particular camels, have been linked to this process of human development and were essential for its success. By establishing trading routes and reusing them over millennia, corridors of gene flow were opened that shaped genetic diversity and structure not only in dromedaries,

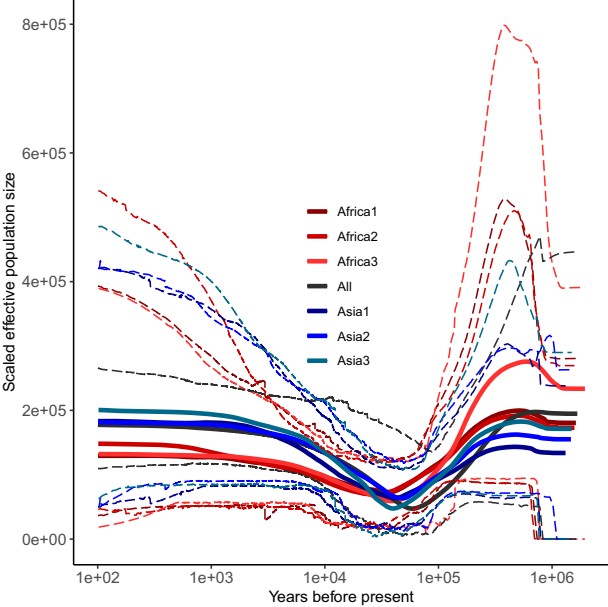

**Fig. 4 Extended Bayesian Skyline Plot (EBSP) for the global, the African and the Asian dromedary population.** EBSP for all dromedaries, with three independent runs per continent (Africa1, Africa2, and Africa3; Asia1, Asia2, and Asia3) and one for the global population (All). Solid lines represent median scaled effective population size and dashed lines represent 95% Highest Posterior Density (HPD) intervals. In each EBSP, 50 random ddRAD loci with at least four but not more than six SNPs across at least 75% individuals were used, and all EBSP runs were calibrated with a RAD locus-specific clock rate to calibrate the time scale. The lowest median effective population size for the different independent runs occurred 58,926 years before present (ybp) for the global population, 37,713, 26,869, and 38,802 ybp for African dromedaries and 41,923, 46,984, and 39,999 ybp for Asian dromedaries, respectively.

but also in Bactrian camels[48], and in many other (livestock) species[49,50]. With a genome-wide dataset we detected patterns of high effective migration in the global dromedary population and revealed pathways of dispersal after domestication, mirroring ancient caravan routes (Fig. 3). While these served as corridors, deserts, and highlands (due to lower population density) represented possible barriers to gene flow. As we estimated an overall mean migration rate, the existing gene flow (below average) between populations along the caravan roads leading through desert regions might have been masked by the high connectivity (above average) of the coastal populations and/or lack of representative samples. Filling in population sampling gaps in future studies will provide a deeper understanding of the gene flow and genetic structure between populations. It will allow a powerful quantification of the magnitude of genetic isolation barriers that may persist.

Dromedaries and Bactrian camels are the most important livestock species in desert areas; their impact on land and water resources for food production is less than that of any other livestock. With increasing desertification and global climate change their importance will grow even more. For this reason, it is essential to understand the demographic history that has shaped modern dromedary populations. The genome-wide diversity present today, which we have characterized in this study, is a result of genetic shuffling due to historic and recent movements along trading routes. This constant mixing might have led to a unique genetic makeup that could make camels more resilient to global environmental changes, and that needs to be preserved.

## Methods

**DNA samples**. We selected a total of 122 dromedary samples from a pool of previously extracted DNA collected during projects supported from the Austrian Science Foundation, FWF P24607-B25 (PI: P.B.) and the European Research Council, EU ENPI CBC MED PROCAMED I.B.1.1/493 (PI: E.C.) under all legal requirements. The samples originated from 18 countries, which were representative of the species' distributional range. In view of potential hybridization between one- and two-humped camels known to occur specifically in Central Asian regions, we included a Bactrian camel to control for introgression and as outgroup for phylogenetic analysis. Detailed information about all samples is provided in Supplementary Data 1.

**Library preparation, sequencing, and initial data filtering**. Library preparation and sequencing was performed at IGA Technology Services (Udine, Italy), to generate genome-wide data from ddRADseq. In silico analysis of the *C. dromedarius* genome assembly (NCBI accession: GCA_000803125.1)[24] highlighted *SphI-BstYI* as the best combination of restriction enzymes able to produce DNA fragments between 400 and 530 bp. ddRAD barcoded libraries were pooled and sequenced on an Illumina HiSeq 2500 in high output mode using 125 bp paired-end reads. Initial raw data analysis as well as SNP calling was performed by IGA Technology Services in-house bioinformatics pipeline. Briefly, all reads were trimmed to 110 bp, and quality controlled reads were aligned to the North African dromedary genome assembly (NCBI accession: GCA_000803125.1) using BWA-MEM[51]. ddRAD reads were then processed with Stacks v.1.35[52]. Out of nearly half a million RAD loci, 88,836 SNPs passed the imposed filtering criteria. The *pstack* module was run with a minimum coverage of 3 reads to call a haplotype, while SNPs were called using a bounded model to highest = 0.5 and alpha = 0.05. *Cstacks* and *sstacks* modules were run with default parameters. Population module was run requiring a minimum genotyping of 75% of individuals to score loci, along with a calling likelihood filtering threshold of −25.

The retrieved raw SNP data were stringently filtered for missing values with PLINK 1.07[53], first excluding individuals with more than 25% missing genotypes (--mind 0.25), next setting a threshold of 0.01 for minor allele frequency (--maf 0.01) and finally, removing SNPs with a missing genotype rate of more than 25% (--geno 0.25). Due to high individual missingness, 13 samples were removed from the dataset as well as 35,013 SNPs after filtering for minor allele frequency and missing genotype rate. To exclude any potentially related individuals, a symmetric identical by state matrix was created with PLINK with a cutoff value of 0.05; one sample from Jordan (JO434) and three from Nigeria (NG852, NG877 and NG887) were removed from further analyses (Supplementary Data 1).

Considering the practice of dromedary and Bactrian camel crossbreeding especially in Central Asian countries[10], we screened for potential hybridization with Bactrian camel present in the dataset. Paired-end ddRAD reads from all 122 dromedaries were simultaneously mapped to either the dromedary (NCBI accession: GCA_000803125.2)[25] or Bactrian camel (NCBI accession: GCF_000767855.1)[23] genomes using BBSplit v. 38.79 (https://sourceforge.net/projects/bbmap/), with the following settings: minratio = 1.0, ambiguous = toss, ambiguous2 = toss. We preprocessed these two genome assemblies with dustmasker v. 1.0[54], and the percent Bactrian camel was estimated as the number of reads that unambiguously mapped to the Bactrian camel genome divided by the total number of unambiguously mapped reads to both dromedary and Bactrian camel multiplied by 100. We removed three individuals from Iran (IR715, IR717, and IR719) and six from Kazakhstan (KZ888, KZ889, KZ890, KZ891, KZ892, and KZ893) (Supplementary Data 2) that had "far-out" values, which are those greater than the third quartile plus the interquartile range multiplied by three[55].

**Genome-wide summary statistics and population structure**. SNPs were tested for Hardy–Weinberg Equilibrium and linkage disequilibrium using VCFTOOLS v0.1.15[56] and as no SNP exceeded a FDR of 0.05, all were retained. For subsequent file conversions, PGDSPIDER version 2.1.1.3[57] was used. The expected ($H_E$) and observed ($H_O$) heterozygosities, AR, and inbreeding coefficients ($F_{IS}$) were calculated with the R package Hierfstat v0.04-22[58,59]. We have used a parametric Welch t test implemented in R v.3.5.1 using the *t.test* function to compare mean $H_E$, $H_O$, and $F_{IS}$ between African and Asian dromedaries. Pairwise $F_{ST}$ and AMOVA[59] were analyzed with the program Arlequin 3.5.2.2[60].

We applied BayeScan 2.1[28] to identify $F_{ST}$ outlier loci putatively under of selection using default settings with a FDR[61] of 0.05. To understand if SNPs putatively detected under selection were linked to significant biological pathways, we screened the respective RAD sequences for genes using the annotation of the new CamDro2 assembly[25] and assessed their protein function with Genecards (http://www.genecards.org). To consider also regions in potential linkage disequilibrium, we included genes 200 kbp upstream and downstream of the SNPs under selection in the analysis.

Individual-based PCA was performed with Adegenet v2.1.1 using the s.class option to represent principal components of known groups. Furthermore we applied ADMIXTURE v1.3[33] to assess ancestry and possible structure among dromedary populations (i.e., countries of origin), using the lowest fivefold cross-validation error to choose the best number of clusters ($K$), from $K = 1$ to $K = 10$. To understand the phylogenetic relationship among individuals, we applied the Neighbor-Net method[62] implemented in SplitsTree4[63], which is a neighbor-joining

algorithm for constructing phylogenetic networks from a genetic (allele sharing) distance matrix created in PLINK.

**Estimating effective migration rates.** We used the software estimated effective migration surfaces (EEMS)[34] to investigate effective migration patterns in the global dromedary population. Based on a stepping-stone model, and assuming that migration is symmetric, EEMS uses Markov Chain Monte Carlo to estimate the diversity and migration parameters and produces maps which represent the posterior mean of effective migration and effective diversity across space. We performed three distinct runs, each consisting of 10 million MCMC iterations, discarding the initial 5 million as burn-in and saving every 49,995 iterations for a grid with 500 demes. All runs reached convergence and results were similar across replicates. The habitat polygon was obtained using Google Maps API v3 Tool (http://www.birdtheme.org/useful/v3tool.html) and results were plotted using the R package rEEMSplots as suggested in Petkova et al.[34].

**Demographic analysis of the global dromedary population.** We assessed the demographic history of the species employing a coalescent-based multi-locus analysis with variable loci using BEAST2 v. 2.5.1[64], setting the Coalescent EBS[40] as a tree prior and following Huson and Bryant[63]. EBS analysis was conducted on the global population where we randomly selected 50 RAD loci containing at least four but not more than six SNPs across at least 75% of the individuals using a custom R script (https://github.com/jelber2/RAD-Scripts/blob/master/RAD_Haplotypes.R)[65]. We repeated the EBS analysis for African and Asian dromedaries separately, but due to slight population structuring, Kenya and *Hadhana* populations were excluded from this analysis. We ran EBS analyses three times per continent, with 50 random RAD loci used in each run. A RAD locus-specific clock rate (per generation) was estimated by calculating the average number of differences between dromedary and Bactrian camel sequences, dividing by the length of the RAD loci, taking the average among SNP classes (number of SNPs per RAD locus from 4 to 6), dividing by the split time between the Bactrian camel and dromedary of ~4.4 million ybp[23], and using a dromedary generation time of 5 years[1]. Each EBS was run for 2,100,000,000 chains using the RAD locus-specific clock rate of 1.809442e-08 to calibrate the time scale.

**Statistics and reproducibility.** Sample size is outlined in Supplementary Data 1. We have used a parametric Welch $t$ test implemented in R v.3.5.1 using the t.test function to compare mean $H_E$, $H_O$, and $F_{IS}$, and data are expressed as mean and SD. Pairwise $F_{ST}$ values were performed including a minimum of three individuals per countries and their significance levels are represented with "+". AMOVA were analyzed with the program Arlequin. $P$ values below 0.05 are considered as statistically significant for all statistical tests in this work. All analyses are reproducible with access to genetic data (see "Data availability").

**Reporting summary.** Further information on research design is available in the Nature Research Reporting Summary linked to this article.

## Data availability
All sequence files (.cram) are deposited at the European Nuceotide Archive with the accession number PRJEB38954 (http://www.ebi.ac.uk/ena/data/view/PRJEB38954). In addition, SNP data (.map and .ped) can be downloaded from Dryad[66] (https://doi.org/10.5061/dryad.kh189322q).

## Code availability
Computer code and scripts for the various analyses are available at Dryad[66] (https://doi.org/10.5061/dryad.kh189322q).

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

## Acknowledgements

We are very grateful to all veterinarian colleagues and camel owners for their agreements to submit sample aliquots for scientific purposes. We thank Steve Smith for lab support and Jukka Corander for general discussion. Samples collection was performed during the Austrian Science Fund (FWF) project number P24607-B25 (to P.B.) and the EU project EU ENPI CBC MED Project PROCAMED. B.1.1/493 (to. E.C.); ddRAD sequencing was performed in the frame of the EU project EU ENPI CBC MED Project PROCAMED. B.1.1/493 (to. E.C.); S.L. and J.P.E. acknowledge funding from the FWF project number P29623-B25 (to P.B.).

## Author contributions

S.L. performed analysis and wrote the first draft of the paper, J.P.E., A.D., D.S., E.T., E.C., and P.B. performed analyses, F.A., N.S., and M.H.B. contributed essential samples, E.C. and P.B. conceived and managed the project and wrote the paper. All authors provided valuable discussions, commented, and approved the final paper.

## Competing interests

All authors declare no competing non-financial interests. Author D.S. is employed by the sequencing company IGA Technologies, Udine, Italy. All other authors declare no competing financial interests.
