## [Peer Review File · Communications Biology]

Reviewers' comments:

Reviewer #1 (Remarks to the Author):

The authors present an interesting paper on genome-wide diversity, gene flow and global migration patterns in dromedaries with findings of current interest, especially in a domestic species insufficiently explored. The paper is well written and conducted. The present work is a useful study and, in my opinion, it could be accepted in the current form.

More in details:

The title of the work almost corresponds to the content of the research. Key words are appropriate. The abstract summarizes the main points of the work. The aim of the research is defined sufficiently, the methodology is clear and complete, the results are clearly presented, the discussion is associated with the results and the conclusion sum up the results of the research.

Introduction section. More data related to origin of dromedary could be referred in the introduction section (e.g. Almathen, F. et al. Ancient and modern DNA reveal dynamics of domestication and cross-continental dispersal of the dromedary)

Line 122-124. This sentence could be deleted. In my opinion the sentence is not supported by the results or should be supported by a reference.

Line 176 and 392. Please use italic style for the gene acronyms and restriction enzyme names.

Supplementary tables. Maybe it could be better to specify ID samples in the footnote.

Table S1. Please specify better if "origin" refers to a sampling area or to previous studies.

Reviewer #2 (Remarks to the Author):

This paper describes the analysis of RADseq data from 122 dromedaries sampled across a wide geographic range. I found the paper very interesting and the analysis mostly sound – it is a genuinely nice new piece of research that will be of importance to people working on domestication. I have, however, a few quibbles and suggestions. In particular, I think the paper requires a bit restructuring (see comments below).

Introduction

There is a lot of superfluous information about history in the first paragraph (some of which I think I not correct such as 4.2k event; see my comment below) yet there not much information about camel genomics in the second. What was done so far in camel?

I am not sure the first sentence is correct – is Fages et al. really saying this? I am also not convinced about the claims later on that there is any evidence that human dispersal per-se was actually facilitated by camels? Trade yes but dispersal?

Line 74: This 3ky before present does not fit the zebu, which seem to arrive in the Near East at the 4.2k event?

Results

Are the 22k SNPs covered in all individual?

I am not sure why the first part of the result is about hybrids while the main question set up in the introduction is whether structure can be detected?

Move background about hybrids in introduction

I do not think that the % of mapped reads to each assembly should be taken as admixture proportions given that a vast majority of reads should map equally well to both.

What was the minimum threshold (proportion of admixture called by ADMIXTURE) to call a sample "hybrid"? There seem to be a long tail in Figure S1 with decreasing levels of admixture. The proportion calculated by ADMIXTURE is suggestive and it is difficult to assess statistical support here. I would recommend that the authors use D-statistics to assess whether the signal is significant especially for these low admixture samples.

I am not sure why the author plot only PC1 and 3 - why not PC2?

It would be great to have a plot for the cross-validation given that it is rare that an analysis of such a wide range of sample indicate that K=1 is optimal

The f3 statistics are extremely underpowered for admixture analysis - most people use those for inferring shared levels of drift. I would remove this analysis from the paper.

Minor

Line 49: medieval, typo.

Just a detail but I would use Results and Discussion / Conclusion heading as there are a lot of interpretation in the result section

Line 239: I do not think that you can say "shared ancestry" here - I would say lack of population structure.

Reviewer #3 (Remarks to the Author):

The current manuscript provides an interesting insight into the fine scale population differentiation in dromedaries across Asia and Africa at genome-wide level. The authors chose double-digest restriction site associated DNA (ddRAD) sequencing on 122 dromedary DNA samples from 18 countries. As a result of ddRAD sequencing 95 dromedaries and 22,721 SNPs were selected for analysis. It was shown, that the genome-wide diversity in modern dromedaries was formed as result of Pleistocene bottleneck and medieval expansions during the rise of the Ottoman empire. It was shown, that the small, locally adapted populations have a great impact in the maintenance of the global diversity of dromedaries and should be the main target for conservation.

The paper is well written and presents an interesting research topic and good ideas. The discussion of produced molecular genetic data in relationship to the ancient and recent history dromedaries has a great value. I have really enjoyed reading the manuscript.

However, I have several questions and recommend the minor revision that should be resolved before its publishing:

- 1) Why only one sample Bactrian camel was included in the analysis? Is it possible to increase the number of Bactrian camel samples?
- 2) The quality-controlled reads were aligned to the North African dromedary genome assembly. It is quite reasonable at study of dromedaries; however, I doubt the correctness of this approach at evaluation of the hybridization pattern with Bactrian camel. To confirm the introgression, authors simultaneously mapped ddRAD reads from putative dromedary-Bactrian camel hybrids to either the

dromedary or Bactrian camel genomes and determined the percentage of unambiguously aligned reads to either dromedary or Bactrian camel genomes. For study of hybridization pattern of dromedaries, I suggest to proceed the ddRAD data without the use of a reference genome and compare the results produced with and without the use of a reference genome.

3) The results of correlation between genetic distances (F_{st}) and geographic distances (Mantel test, Fig. S2) are questionable. Authors estimated the geographic distances between populations based on their belonging to the country. I suppose, it is not quite correct. I suggest to exclude the Mantel test results from the manuscript.

Dr. Brooke LaFlamme, Chief Editor, Communications Biology:

Your manuscript entitled "Genome-wide diversity and global migration patterns in dromedaries follow ancient caravan routes" has now been seen by 3 referees. Thank you for your patience with the longer than usual review process. You will see from the referees' comments below that they find your work of considerable interest and have only minor requests for revision. We are interested in the possibility of publishing your study in Communications Biology, but would like to consider your response to these concerns in the form of a revised manuscript before we make a final decision on publication.

We therefore invite you to revise and resubmit your manuscript, taking into account the points raised. Please highlight all changes in the manuscript text file.

We appreciate all reviewers' comments, which have helped us to improve the quality of our manuscript. We have carefully revised and responded to all reviewers' suggestions.

Referee expertise:

Referee #1: Genetic analysis in domesticated animals, population structure

Referee #2: animal population genomics, domestication

Referee #3: animal population genomics, domestication

Reviewers' comments:

Reviewer #1 (Remarks to the Author):

The authors present an interesting paper on genome-wide diversity, gene flow and global migration patterns in dromedaries with findings of current interest, especially in a domestic species insufficiently explored. The paper is well written and conducted. The present work is a useful study and, in my opinion, it could be accepted in the current form.

More in details:

The title of the work almost corresponds to the content of the research. Key words are appropriate. The abstract summarizes the main points of the work. The aim of the research is defined sufficiently, the methodology is clear and complete, the results are clearly presented, the discussion is associated with the results and the conclusion sum up the results of the research.

We very much appreciate the reviewer's opinion.

Introduction section. More data related to origin of dromedary could be referred in the introduction section (e.g. Almathen, F. et al. Ancient and modern DNA reveal dynamics of domestication and cross-continental dispersal of the dromedary)

We thank the reviewer for the suggestion and changed the introduction to emphasize what has been done so far (192-198). We have also included now two sentences about the origin of Old World camels (180-84). Specifically to the Almathen et al 2016 suggestion, we have cited the paper thirteen times throughout the manuscript and considering self-citation we should probably not include it more often.

Line 122-124. This sentence could be deleted. In my opinion the sentence is not supported by the results or should be supported by a reference.

We have changed it according to the suggestion.

Line 176 and 392. Please use italic style for the gene acronyms and restriction enzyme names.

We have changed all gene acronyms and restriction enzyme names throughout the manuscript according to the suggestion.

Supplementary tables. Maybe it could be better to specify ID samples in the footnote.

We have changed it according to the suggestion.

Table S1. Please specify better if “origin” refers to a sampling area or to previous studies.

We have changed it according to the suggestion.

Reviewer #2 (Remarks to the Author):

This paper describes the analysis of RADseq data from 122 dromedaries sampled across a wide geographic range. I found the paper very interesting and the analysis mostly sound – it is a genuinely nice new piece of research that will be of importance to people working on domestication. I have, however, a few quibbles and suggestions. In particular, I think the paper requires a bit restructuring (see comments below).

Introduction

There is a lot of superfluous information about history in the first paragraph (some of which I think I not correct such as 4.2k event; see my comment below) yet there not much information about camel genomics in the second. What was done so far in camel?

We thank the reviewer for the suggestion and changed the final part of the introduction to emphasize which genomic studies have been performed in camels (194-98).

I am not sure the first sentence is correct – is Fages et al. really saying this?

We thank Reviewer 2 for the comment. Fages et al. 2019 actually mentions that

„Horses provided humans with the first opportunity to spread genes, diseases, and culture well above their own speed“ and „Horses also revolutionized warfare, pulling chariots at full speed in the Bronze Age, providing the foundation for mounted battle in the early Iron Age, and facilitating the spread of cavalry during Antiquity“.

We added another citation (Herrera et al. 2018) stating „Until the Industrial Revolution and the invention of the railroads, large mammals served as the main mode of transportation. Before animal domestication, humans transported goods and people on their own backs. Animals such as donkeys, horses, and llamas made it possible to carry heavy goods over longer distances with minimal resources.“

We think that our first sentence captures the importance of large domestic animals for human migration and transportation, we consider it to be correct and we would prefer to keep it in the manuscript (157-58).

I am also not convinced about the claims later on that there is any evidence that human dispersal per-se was actually facilitated by camels? Trade yes but dispersal?

Considering the reviewer’s comment, we changed the sentence to:

„One of our latest domesticates (~3000-4000 years before present (ybp)), the dromedary (*Camelus dromedarius*), has a special position in human migration and trading. “ (158-60).

Line 74: This 3ky before present does not fit the zebu, which seem to arrive in the Near East at the 4.2k event?

We can only refer to the dates given in the respective literature, which we have highlighted below (Edwards et al. 2007). If the reviewer would kindly provide the literature referring to the 4.2k event, we would be able to discuss and integrate it in our text.

In Taurine and zebu admixture in Near Eastern cattle: a comparison of mitochondrial, autosomal and Y-chromosomal data (Edwards et al 2007)

“...This scenario is supported by analyses of linked microsatellite markers on the bovine X chromosome, which indicate that the age of zebu–taurine admixture in the Near East is relatively ancient (Freeman et al. 2006b). It is currently difficult to date the onset or maxim of this influx; however, archaeological evidence suggests the presence of zebu cattle in Jordan dating to 3400 BP (Clason 1978). It is therefore possible that the significant cline of zebu influence across the Near East reflects the influence of ancient trade links, such as the silk route. In addition, the mid-Holocene (8000 to 3000 BP) was a time of substantial climatic variability (Sandweiss et al. 1999), and there is environmental evidence that Mesopotamia suffered a prolonged drought around 4000–3000 BP (Neumann & Parpola 1987). Consequently, it has been suggested that ancient herders may have introduced significant numbers of arid-adapted zebu populations into the Near East at this time (Matthews 2002). In

addition, Loftus et al. (1999) have speculated that the Islamic civilizations in the region may have favored zebu-type cattle because of their links with Islamic culture and religion.”

Results

Are the 22k SNPs covered in all individual?

In PLINK we applied an individual genotype filter of `--geno 0.25`, which means that only those SNP were retained, which were present in at least 75% of the individuals.

I am not sure why the first part of the result is about hybrids while the main question set up in the introduction is whether structure can be detected?

First of all, the amount of hybridization observed in our data was not expected, as all samples were given to us as purebred dromedaries. However, based on the history of Old World camels in Central Asian countries, the practice of hybridization is widespread, we had to consider, and test for it, before a proper assessment of population structure in the global dromedary population was possible. Therefore, we decided to present the result in the (logic) order of the workflow, first removing the hybrids from further analysis and then describing population structure. As the presentation of results was not mentioned by any of the other reviewers, nor by the editor and it seems logic to us, we would prefer to keep it like it is now in the manuscript.

Move background about hybrids in introduction

We thank for the suggestion and moved parts of the background from Results to Introduction section (169–71).

I do not think that the % of mapped reads to each assembly should be taken as admixture proportions given that a vast majority of reads should map equally well to both.

We thank for the comment. We do agree that there is high proportion of reads that map ambiguously (i.e., map equally well to both dromedary and Bactrian camel genomes). We actually report what percentage of unambiguously mapped reads belongs to each of the genomes. We have changed the text to explain that this is an alternative method for estimating hybridization/admixture proportions and that is not necessarily indicative of actual proportions of shared ancestry (1441-443). We use this method to complement the other methods we employed to predict whether an individual was or was not a hybrid.

What was the minimum threshold (proportion of admixture called by ADMXITURE) to call a

sample "hybrid"? There seem to be a long tail in Figure S1 with decreasing levels of admixture. The proportion calculated by ADMIXTURE is suggestive and it is difficult to assess statistical support here. I would recommend that the authors use D-statistics to assess whether the signal is significant especially for these low admixture samples.

We used different methods to complement each other at predicting whether an individual was or was not a hybrid. According to our understanding we cannot use D-statistics (ABBA-BABA) for this purpose, as at least one outgroup to the Old World camels, in this case New World camels, would be needed, which were not ddRAD sequenced.

I am not sure why the author plot only PC1 and 3 - why not PC2?

Plotting PC1 and PC3 is giving more visual information to the reader, and PC1 and PC2 is shown Figure S4.

It would be great to have a plot for the cross-validation given that it is rare that an analysis of such a wide range of sample indicate that $K=1$ is optimal

We agree with the reviewer's comment and include it as "Figure S2".

The f_3 statistics are extremely underpowered for admixture analysis - most people use those for inferring shared levels of drift. I would remove this analysis from the paper.

We agree and have removed the f_3 statistics according to the suggestion.

Minor

Line 49: medieval, typo.

We have changed it according to the suggestion.

Just a detail but I would use Results and Discussion / Conclusion heading as there are a lot of interpretation in the result section

We changed the headings following the reviewer's suggestion. Originally, we followed the guidelines of Nature journals (<https://www.nature.com/ncomms/submit/article>), but we agree with the reviewer. However, we will leave it up to the technical editor of Communications Biology to decide on this topic.

Line 239: I do not think that you can say "shared ancestry" here - I would say lack of population structure.

We have changed it according to the suggestion.

Reviewer #3 (Remarks to the Author):

The current manuscript provides an interesting insight into the fine scale population differentiation in dromedaries across Asia and Africa at genome-wide level. The authors chose double-digest restriction site associated DNA (ddRAD) sequencing on 122 dromedary DNA samples from 18 countries. As a result of ddRAD sequencing 95 dromedaries and 22,721 SNPs were selected for analysis. It was shown, that the genome-wide diversity in modern dromedaries was formed as result of Pleistocene bottleneck and medieaval expansions during the rise of the Ottoman empire. It was shown, that the small, locally adapted populations have a great impact in the maintenance of the global diversity of dromedaries and should be the main target for conservation.

The paper is well written and presents an interesting research topic and good ideas. The discussion of produced molecular genetic data in relationship to the ancient and recent history dromedaries has a great value. I have really enjoyed reading the manuscript.

However, I have several questions and recommend the minor revision that should be resolved before its publishing:

1) Why only one sample Bactrian camel was included in the analysis? Is it possible to increase the number of Bactrian camel samples?

From the design of our study we were interested in ddRAD sequencing as many dromedaries form a global distribution range as possible (allowed by our budget). However, being aware that hybridization between dromedaries and Bactrian camels occurs, we included one Bactrian camel to test and control for introgression. Certainly, it would be interesting to perform a similar study on Bactrian camels, however, we are lacking the global sample set and to our understanding, another group of Chinese scientists is already working on it.

2) The quality-controlled reads were aligned to the North African dromedary genome assembly. It is quite reasonable at study of dromedaries; however, I doubt the correctness of this approach at evaluation of the hybridization pattern with Bactrian camel. To confirm the introgression, authors simultaneously mapped ddRAD reads from putative dromedary-Bactrian camel hybrids to either the dromedary or Bactrian camel genomes and determined the percentage of unambiguously aligned reads to either dromedary or Bactrian camel genomes. For study of hybridization pattern of dromedaries, I suggest to proceed the ddRAD data without the use of a reference genome and compare the results produced with and without the use of a reference genome.

We appreciate the suggestion although we disagree to *de novo* assemble the ddRAD data. *De novo* assembly generates far too many false positives regarding retained RAD loci and hence SNPs. *De novo* assembly does allow for overcoming reference bias (i.e., calling variation in regions that might not be in the reference) but at the cost of a higher false positive rate whereby for example alternate haplotypes/alleles are not assembled together and are treated as separate loci that confound sensitivity of SNP and genotype calling.

3) The results of correlation between genetic distances (F_{st}) and geographic distances (Mantel test, Fig. S2) are questionable. Authors estimated the geographic distances between populations based on their belonging to the country. I suppose, it is not quite correct. I suggest to exclude the Mantel test results from the manuscript.

We agree and have removed it according to the suggestion.

Reviewers' comments:

Reviewer #2 (Remarks to the Author):

Overall the authors have addressed/taken on board very few of my comments.

I still think the first sentence is an overstatement. Firstly, there were people on all continents thousands of years before the domestication of horse or camel (50,000 years before in Australia!). Secondly, Fages et al. is a genomic paper and does not seem at all an adequate citation for this statement it is as a best secondary citation.

The literature on zebu is very outdated – please see this paper: [10.1126/science.aav1002](https://doi.org/10.1126/science.aav1002) published in Science last year.

Please include the sentence about 75% coverage in the manuscript.

The arguments about the lack of outgroup for the D-statistics is not convincing. Are there not any whole genome data available for New World Camel (e.g. there is a genome of an Alpaca available)? Could you just not align this to the same reference genome and pull the genotype of this outgroup at the same SNP loci? I think this is a key analysis to do here. One simple way to do it would be to chop the genome of the Alpaca (or any other outgroup) into 500bp – align those to the reference genome used in this study and call the allele (as ancestral) at the same RAD loci.

About PC1-2 – this is not really good practice. You should at least show the two main axes in a single graph rather than selecting which result fit best your narrative...

Reviewer #3 (Remarks to the Author):

The authors took my comments into account or provided reasonable explanations. I believe that the manuscript can be published in its current form.

COMMSBIO-19-1902A "Genome-wide diversity and global migration patterns in dromedaries follow ancient caravan routes"

Answer to Reviewers' comments: Reviewer #2 (Remarks to the Author):

1) Overall the authors have addressed/taken on board very few of my comments.

We are very sorry that Reviewer 2 felt this way, it was not at all our intention. We actually addressed each of the 14 comments, and we fully implemented nine of them.

2) I still think the first sentence is an overstatement. Firstly, there were people on all continents thousands of years before the domestication of horse or camel (50,000 years before in Australia!). Secondly, Fages et al. is a genomic paper and does not seem at all an adequate citation for this statement it is as a best secondary citation.

Following the reviewer's comment, we have eliminated the first sentence of the introduction.

3) The literature on zebu is very outdated – please see this paper: 10.1126/science.aav1002 published in Science last year.

We thank the reviewer for the additional literature; based on the present and previous statement, we have eliminated the complete sentence (I73-74).

4) Please include the sentence about 75% coverage in the manuscript.

We have changed the sentence according to the suggestion (I114-115).

5) The arguments about the lack of outgroup for the D-statistics is not convincing. Are there not any whole genome data available for New World Camel (e.g. there is a genome of an Alpaca available)? Could you just not align this to the same reference genome and pull the genotype of this outgroup at the same SNP loci? I think this is a key analysis to do here. One simple way to do it would be to chop the genome of the Alapaca (or any other outgroup) into 500bp – align those to the reference genome used in this study and call the allele (as ancestral) at the same RAD loci.

We understand the concerns of Reviewer 2 regarding a valid threshold for identifying putative hybrids and we would like to draw the reviewer's and Editor's attention to the original request of Reviewer 2:

“What was the minimum threshold (proportion of admixture called by ADMXITURE) to call a sample "hybrid"? There seem to be a long tail in Figure S1 with decreasing levels of admixture. The proportion calculated by ADMIXTURE is suggestive and it is difficult to assess statistical support here. I would recommend that the authors use D-statistics to assess whether the signal is significant especially for these low admixture samples.”

To follow Reviewer 2's request, we mapped paired-end ddRAD reads from all 122 dromedaries simultaneously to either the dromedary (NCBI accession: GCA_000803125.2) or Bactrian camel genomes (NCBI accession: GCF_000767855.1)

(I425-436) and made a boxplot (new Figure 1, see below) showing the percentage of reads mapping to the Bactrian camel genome. We used the 3rd quartile + interquartile range of all values x 3 to define a cut-off and identified outliers, which correspond to the nine samples previously considered as putative hybrids. We hope with this additional analysis, we could address the request of Reviewer 2 to define the threshold of calling a sample “putative hybrid”.

Fig. 1. Putative hybrids detected. Boxplot of 122 dromedary samples based on each sample’s estimated percent of reads mapping to the Bactrian camel genome. The thick black line represents the boxplot of all percent values (Table S2), which is heavily compressed due to the high similarity of all values in pure-bred dromedaries. The nine outliers represent samples that we concluded as putative hybrids based on possessing “far out” values (Tukey 1977). The blue line is the upper extreme value that we used as a cutoff for hybrids calculated as: 3rd quartile + interquartile range x 3. Note that we did not plot the lower “far out” values (i.e., values below the solid black line) as our goal was to predict putative hybrids that we defined as upper “far out” values.

Regarding the suggestion of applying D-Statistics in our analysis, unfortunately our possibilities are limited. We really would like to implement this method, and we understand that it is possible to even use single genomes. However, we do not know how it could be applied on dromedary and Bactrian camels, considering that the underlying assumption of D-statistics is to detect gene flow between two in-groups that are not sister species, based on data from four taxa with an established phylogeny.

We refer to Zheng Y, Janke A. *Gene flow analysis method, the D-statistic, is robust in a wide parameter space. BMC Bioinformatics. 2018 Jan 8;19(1):10. doi: 10.1186/s12859-017-2002-4.*

The D-statistic (see Methods for formula) is used for a group of four taxa with an established phylogeny (Fig. 1) to detect gene flow between two ingroups that are not sister species (in this case, H2 and H3). The value of D is affected by a number of parameters; a) fraction of gene flow (f), b) divergence times, c) time of gene flow

We asked the corresponding author of Zheng & Janke (2018) for further details on the methodology. Below, please find Dr. Zheng's reply; the full correspondence is available upon request.

from: yzheng2@uni-koeln.de
to: Pamela Burger <pamela.burger@vetmeduni.ac.at>
date: 1 Apr 2020, 21:31
subject: Re: Application of D-Statistics on ddRAD SNP data
Signed by: uni-koeln.de

Dear Dr. Burger,

You are correct that D-stats (and related f-stats) are not able to detect gene flow between sister taxa. The reason is that D- and f-stats are looking for signals that are inconsistent with the phylogeny; but gene flow between sister taxa does not produce such inconsistency. [...]

We would like to emphasize that we took great efforts to consider Reviewer 2's suggestion, and we politely want to draw the reviewer's attention to the fact that it was not the main goal of our paper to describe gene flow and hybridization between one-

and two-humped camels. We are aware of hybridization in regions where both camel species co-exist, and we simply intended to remove potential hybrids from the dataset for downstream analysis. Thus, it was not a key point of this manuscript, and we hope that our arguments explaining the limitations of our data set regarding the D-Statistics method are well received by Reviewer 2.

6) About PC1-2 – this is not really good practice. You should at least show the two main axes in a single graph rather than selecting which result fit best your narrative...

Following the reviewer's suggestion we have moved the graph showing PC1-2 (previously Supplementary Figure 4) into the main manuscript and combined it with PC1-3 into a new Figure 2.

Reviewers' comments:

Reviewer #2 (Remarks to the Author):

I do not understand why the authors are doubling down on their mistake, quoting an email taken out of context from what was likely a very unspecific question (sister taxa is a relative term, plants and animals are sister taxa relative to bacterias...), rather than engaging with the constructive suggestions here. The authors meant that you would not be able to test for admixture between two genomes if all you have are the two genomes.

The authors completely misunderstood D statistics, it would have been useful to read the paper carefully or ask the author of the paper they've emailed (or myself through this review process) about the specific issue they did not understand.

In their Figure 1 (taken from the paper they quoted) in their response to my comment - H1 and H2 can be any combination of dromedary genome and H3 is a Bactrian camel, H4 is an outgroup. If no admixture there should be the same number of ABBA (sites where H1 matches H3) and BABA (sites where H2 matches H3), in this case the $D=0$. If there are more ABBA than BABA or vice-versa D can be < 0 or > 0 , which is suggestive of admixture. Significance can be computed using jackknifing - all this is implemented in the ADMIXTOOLS package - <https://github.com/DReichLab/AdmixTools/blob/master/README.Dstatistics>

So here the authors could take a H1= a dromedary that shows no sign of admixture and H2= a dromedary that shows sign of admixture (using their very rough estimation using reads) and see whether this is indeed significant. This will tell them whether admixture between bactrian camel and dromedary (H2) has taken place since the common ancestor of H1/H2.

Answer to Reviewer 2 and overview of changes throughout the manuscript

Reviewers' comments:

Reviewer #2 (Remarks to the Author):

I do not understand why the authors are doubling down on their mistake, quoting an email taken out of context from what was likely a very unspecific question (sister taxa is a relative term, plants and animals are sister taxa relative to bacterias...), rather than engaging with the constructive suggestions here. The authors meant that you would not be able to test for admixture between two genomes if all you have are the two genomes.

The authors completely misunderstood D statistics, it would have been useful to read the paper carefully or ask the author of the paper they've emailed (or myself through this review process) about the specific issue they did not understand.

In their Figure 1 (taken from the paper they quoted) in their response to my comment - H1 and H2 can be any combination of dromedary genome and H3 is a Bactrian camel, H4 is an outgroup. If no admixture there should be the same number of ABBA (sites where H1 matches H3) and BABA (sites where H2 matches H3), in this case the $D=0$. If there are more ABBA than BABA or vice-versa D can be < 0 or > 0 , which is suggestive of admixture. Significance can be computed using jackknifing - all this is implemented in the ADMIXTOOLS package -

<https://github.com/DReichLab/AdmixTools/blob/master/README.Dstatistics>

So here the authors could take a H1= a dromedary that shows no sign of admixture and H2= a dromedary that shows sign of admixture (using their very rough estimation using reads) and see whether this is indeed significant. This will tell them whether admixture between bactrian camel and dromedary (H2) has taken place since the common ancestor of H1/H2.

Answer to Reviewer 2

We very much appreciate the additional explanatory comments of Reviewer 2, which we certainly will take into consideration for a future manuscript specifically about hybridization between the two camel species (in preparation). However, for the current manuscript we prefer to follow the suggestion of the Editor to remove the hybridization section from the paper.

Overview of changes throughout the manuscript

Introduction

Line 64: We deleted "anthropogenic".

Results / discussion

Lines 122-147: We deleted the entire paragraph "Anthropogenic interspecific hybridisation between Bactrian camels and dromedaries from Iran and Kazakhstan".

Lines 110-120: Instead, we kept two sentences at the beginning of the results to explain why the individuals were removed, next to other stringent filtering steps:

We performed double-digest restriction site associated DNA (ddRAD) sequencing on 122 dromedary DNA samples from 18 countries (Table S1) representative of the species distribution range. We included one Bactrian camel to test for potential interspecific hybridisation, as this is a widespread practice in Central Asia until today that might have even started in pre-Roman times ¹¹. Higher numbers of reads mapping to the Bactrian camel were detected in three individuals from Iran and in six from Kazakhstan (see methods), and we decided to remove these samples from downstream analysis due to potential introgression from Bactrian camel (Table S2). After stringent filtering for genotype and individual missingness, minor allele frequency and relatedness, the final dataset consisted of 95 dromedaries and 22,721 SNPs present in at least 75% of the individuals.

Figure 1 and Table S2: We also removed Figure 1 – "Putative hybrids detected" and deleted the last column of Table S2, which indicated putative hybrids – instead we added an asterisk to those individuals that were eliminated due to high percentage of reads mapping to the Bactrian camel genome.

Conclusion

Lines 381-382: We deleted part of a sentence mentioning the admixture between species:

We detected genetic admixture not only across continental populations (Asia and Africa), but also between species (dromedaries and Bactrian camels), which highlights the strong anthropogenic influence on these animals.

reads now:

We detected genetic admixture across continental populations (Asia and Africa), which highlights the strong anthropogenic influence on these animals

Methods

Line 461-474: The header *Hybridization between dromedaries and Bactrian camels* was removed and the consecutive paragraph was re-written and integrated in the section *Library preparation, sequencing and initial data filtering*

Considering the practice of dromedary and Bactrian camel cross-breeding especially in Central Asian countries¹⁰, we ***tested for potential hybridization with Bactrian camel present in the dataset***. Paired-end ddRAD reads from all 122 dromedaries were simultaneously mapped to either the dromedary (NCBI accession: GCA_000803125.2)²⁵ or Bactrian camel (NCBI accession: GCF_000767855.1)²³ genomes using BBSplit v. 38.79 (<https://sourceforge.net/projects/bbmap/>), with the following settings: *minratio=1.0 ambiguous=toss ambiguous2=toss*. We pre-processed these two genome assemblies with *dustmasker v. 1.0*⁵³, and the percent Bactrian camel was estimated as the number of reads that unambiguously mapped to the Bactrian camel genome divided by the total number of unambiguously mapped reads to both dromedary and Bactrian camel multiplied by 100. We removed ***three individuals from Iran (IR715, IR717, IR719) and six from Kazakhstan (KZ888, KZ889, KZ890, KZ891, KZ892, KZ893)*** (Table S2) that had “far out” values, which are those greater than the third quartile plus the interquartile range multiplied by three⁵⁴.

Furthermore, we deleted the Admixture analysis between dromedaries and Bactrian camel and the respective Figure S1.

Finally, all Figure and Table numbers were adjusted throughout the manuscript.